# High energy-density and reversibility of iron fluoride cathode enabled via an intercalation-extrusion reaction

Xiulin Fan[1], Enyuan Hu [2], Xiao Ji[1], Yizhou Zhu[3], Fudong Han[1], Sooyeon Hwang[2], Jue Liu[4], Seongmin Bak[2], Zhaohui Ma [1], Tao Gao[1], Sz-Chian Liou[5], Jianming Bai[2], Xiao-Qing Yang[2], Yifei Mo [3], Kang Xu[6], Dong Su [2] & Chunsheng Wang [1]

Iron fluoride, an intercalation-conversion cathode for lithium ion batteries, promises a high theoretical energy density of 1922 Wh kg$^{-1}$. However, poor electrochemical reversibility due to repeated breaking/reformation of metal fluoride bonds poses a grand challenge for its practical application. Here we report that both a high reversibility over 1000 cycles and a high capacity of 420 mAh g$^{-1}$ can be realized by concerted doping of cobalt and oxygen into iron fluoride. In the doped nanorods, an energy density of ~1000 Wh kg$^{-1}$ with a decay rate of 0.03% per cycle is achieved. The anion's and cation's co-substitutions thermodynamically reduce conversion reaction potential and shift the reaction from less-reversible intercalation-conversion reaction in iron fluoride to a highly reversible intercalation-extrusion reaction in doped material. The co-substitution strategy to tune the thermodynamic features of the reactions could be extended to other high energy conversion materials for improved performance.

[1] Department of Chemical and Biomolecular Engineering, University of Maryland, College Park, MD 20742, USA. [2] Brookhaven National Laboratory, Upton, NY 11973, USA. [3] Department of Materials Science and Engineering, University of Maryland, College Park, MD 20742, USA. [4] Chemical and Engineering Materials Division, Oak Ridge National Laboratory, Oak Ridge, TN 37831, USA. [5] Maryland Nanocenter, University of Maryland, College Park, MD 20742, USA. [6] Electrochemistry Branch, Power and Energy Division Sensor and Electron Devices Directorate, U.S. Army Research Laboratory, Adelphi, MD 20783, USA. These authors contributed equally: Xiulin Fan, Enyuan Hu. Correspondence and requests for materials should be addressed to D.S. (email: dsu@bnl.gov) or to C.W. (email: cswang@umd.edu)

L ithium ion batteries (LIBs) have dominated portable electronics, and are penetrating the markets of electric vehicles. Current LIB cathodes such as $LiCoO_2$, $LiFePO_4$, or $LiNi_xMn_yCo_{1-x-y}O_2$ are exclusively based on intercalation mechanism, which involves topotactic intercalation/deintercalation of $Li^+$ in a host lattice. However, these cathodes have specific capacities ranged only from 140 to 200 mAh g$^{-1}$, which limit their energy densities[1,2]. Alternatively, certain metallic oxide, sulfide, or halide compounds can experience conversion reactions with $Li^+$ by accepting multiple electrons per formula, delivering much higher capacities. These conversion reaction compounds, represented as $MX_y$ (M = transition metals, X = N, F, O, S, etc.), are converted to a nanocomposite containing reduced $M^0$ and $Li_zX$ after discharge (Eq. 1)[3]:

$$M^{y+}X^{z-}_{y/z} + yLi^+ + ye^- \longleftrightarrow M^0 + y/zLi_zX \quad (1)$$

Among them, iron trifluoride ($FeF_3$), cheap and environmentally friendly, appears to be unique as a cathode candidate with a total theoretical energy density of 1922 Wh kg$^{-1}$ and a relatively high working potential due to its ionic nature[4]. Its lithiation process goes through two successive intercalation and conversion reactions. In the intercalation reaction at ~3.0 V, $FeF_3$ is lithiated into $LiFeF_3$ providing a capacity of 237 mAh g$^{-1}$. With further lithiation to below 2.0 V, the $LiFeF_3$ will decompose into Fe and LiF through a conversion reaction providing another 475 mAh g$^{-1}$ (Eq. 2)[5].

$$FeF_3 \overset{Li^+ + e^- \text{ intercalation}}{\underset{237\,mAh\,g^{-1}}{\longleftrightarrow}} LiFeF_3 \overset{2Li^+ + 2e^- \text{ conversion}}{\underset{475\,mAh\,g^{-1}}{\longleftrightarrow}} 3LiF + Fe \quad (2)$$

Although extensive researches have been conducted, only limited performance improvement was achieved due to several intrinsic issues of conversion reaction: (1) a severe voltage hysteresis (~1.3 V) between the lithiation and delithiation processes, thus a low round-trip energy efficiency of <60% was always observed[6–8], due to the slow phase separation and repeated breaking/reforming of Fe–F bonds in each conversion reaction cycle[9,10]; (2) sluggish conversion reaction kinetics, which is caused by both poor electronic conduction, low ion diffusivity in $FeF_3$[8], and the slow conversion phase transition, leads to extremely poor rate capabilities[7,11–13]; (3) the aggregation and continuous coarsening of Fe nanoparticles[13,14] during repeated conversion reaction cycles, and sustained reactions of Fe with electrolytes[15,16], result in the rapid capacity decay during cycling[4,16–30]. Partial substitution of fluorine with oxygen ($FeO_xF_{2-x}$ (0.4 < x < 0.7)) enable the formation of an intermediate rocksalt phase through a reversible extrusion reaction before the conversion reaction of rocksalt phase. During cell discharge, the incoming $Li^+$ enter the parent phase and prompt the precipitation of metal and LiF phase. On recharge, these precipitated phases go into the parent phase with the $Li^+$ pumped out. Here the parent phase is the defected rocksalt phase. The lithiation of $FeO_xF_{2-x}$ experiences intercalation-extrusion-conversion reaction pathway. Since the less-reversible conversion reaction was partially replaced by a highly reversible extrusion reaction in $FeO_xF_{2-x}$ (0.4 < x < 0.7), electrochemical performance was significantly enhanced (Eq. 3)[6,31–37]. However, the reaction kinetics, cycle life, and round-trip efficiency of $FeO_xF_{2-x}$ (0.4 < x < 0.7) still are far less satisfactory when compared to intercalation cathodes due to the existence of less-reversible conversion reaction.

$$FeOF \overset{Li^+ + e^- \text{ intercalation}}{\longleftrightarrow} LiFeOF$$
$$\text{(rutile)} \qquad\qquad \text{(rutile)}$$
$$\overset{(x+y-1)Li^+ + (x+y-1)e^- \text{ Extrusion I}}{\longleftrightarrow} Li_xFeOF_{1-y} + yLiF$$
$$\text{(rocksalt)}$$
$$\overset{(1.5-x-y)Li^+ + (1.5-x-y)e^- \text{ Extrusion II}}{\longleftrightarrow} 0.5 \underset{\text{(rocksalt)}}{LiFeO_2} + LiF + 0.5Fe$$
$$\overset{1.5Li^+ + 1.5e^- \text{ Conversion}}{\longleftrightarrow} Fe + LiF + Li_2O \qquad (3)$$

In recent years, the critical issues of high potential hysteresis (>1 V), poor rate capability, and limited cycling stability of conversion electrodes have been believed to be intrinsic nature of the conversion reaction chemistry, and the hope of using conversion reaction materials in the next-generation lithium batteries waned. However, it will be plausible to suppress the conversion reaction by extending the capacity of highly reversible intercalation-extrusion reaction, thus achieving both high capacity and long cycle life.

In this work, we report that such a high-performance $Fe_{0.9}Co_{0.1}OF$ cathode with high energy density of ~1000 Wh kg$^{-1}$ and long cycle life of 1000 cycles can be realized by a cost-effective and simple strategy of concerted doping Co/O in $FeF_3$. At a charge/discharge current of 500 mA g$^{-1}$, the $Fe_{0.9}Co_{0.1}OF$ can deliver a capacity of 350 mAh g$^{-1}$ for 1000 cycles. The first-principles calculations and comprehensive characterizations revealed that the less-reversible conversion reaction was entirely prohibited by reducing the potential $f$ and the reversibility of the extrusion reaction II was significantly enhanced by formation of non-stoichiometric rocksalt phase with only <5% of Fe(Co) phase in lithiated $Fe_{0.9}Co_{0.1}OF$ after 100 cycles (Eq. 4), while 50% of Fe was reported in lithiated $FeO_xF_{2-x}$ (0.4 < x < 0.7)[38]. The thermodynamic capacity reduction due to the elimination of conversion reaction was compensated by kinetic capacity increase due to the enhanced reaction kinetics of extrusion II through formation of defect-rich (especially O vacancies) rocksalt phase. The highly reversible structural transition in the intercalation and extrusion reactions after co-substitution significantly improved performance of $Fe_{0.9}Co_{0.1}OF$, which provides an alternative solution to the similar challenges faced by other conversion reaction materials.

$$Fe(Co)OF \overset{Li^+ + e^- \text{ Intercalation}}{\longleftrightarrow} LiFe(Co)OF$$
$$\text{(rutile)} \qquad\qquad \text{(rutile)}$$
$$\overset{(x+y-1)Li^+ + (x+y-1)e^- \text{ Extrusion I}}{\longleftrightarrow} Li_xFe(Co)OF_{1-y} + yLiF$$
$$\text{(rocksalt)}$$
$$\overset{(1.5-x-y)Li^+ + (1.5-x-y)e^- \text{ Extrusion II}}{\longleftrightarrow} \underset{\text{(defected rocksalt)}}{Li_{1-\frac{z}{2}}Fe(Co)O_{2-z}} \qquad (4)$$
$$\frac{1}{2-z}$$
$$+ LiF + (1 - \frac{1}{2-z})Fe(Co)$$

## Results

**Synthesis and characterization.** $Fe_{0.9}Co_{0.1}OF$ and FeOF were synthesized using a solvothermal method, while $FeF_3$ was prepared by ball milling the commercial $FeF_3$ for 6 h (Supplementary Fig. 1). $FeF_3$, FeOF, and $Fe_{0.9}Co_{0.1}OF$ powders show distinct colors: $FeF_3$ is light green; FeOF appears brown; while $Fe_{0.9}Co_{0.1}OF$ is dark brown (Supplementary Fig. 2). This darkening trend indicates the change of the bandgap after doping, which is verified by the density functional theory (DFT) calculations (Supplementary Fig. 3). The crystal structures of FeOF and $Fe_{0.9}Co_{0.1}OF$ were

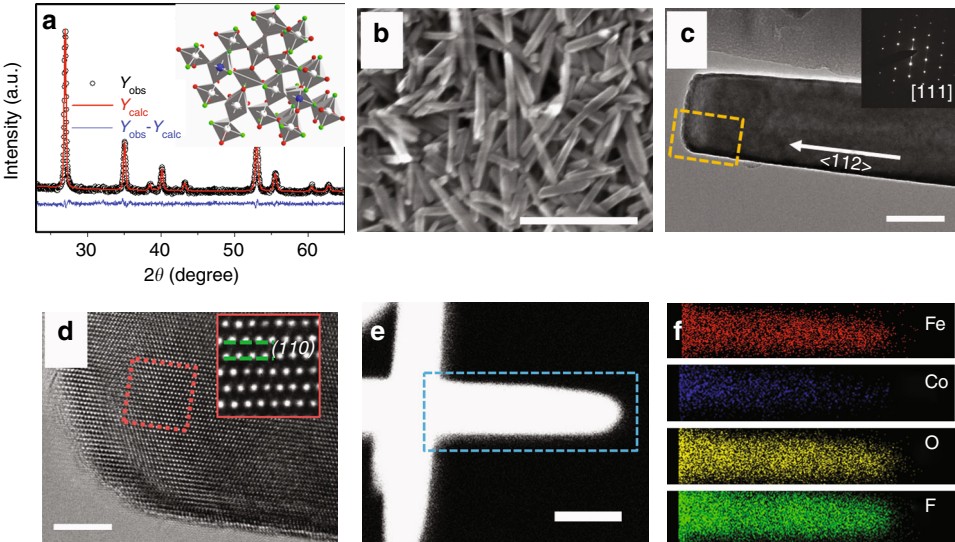

**Fig. 1** Structural and morphological characterization of $Fe_{0.9}Co_{0.1}OF$. **a** Rietveld refinement showing data points ($Y_{obs}$, black circles), calculated profile ($Y_{calc}$, red line), and difference profile (blue line), as indicated. Refined parameters: $a = 4.68(6)$ Å, $c = 3.06(6)$ Å, which are slightly larger than those of FeOF ($a = 4.647$ Å, $c = 3.048$ Å). **b** SEM images of the $Fe_{0.9}Co_{0.1}OF$ nanorods; **c**, **d** TEM and HRTEM images of $Fe_{0.9}Co_{0.1}OF$; **e**, **f** STEM-HAADF image of $Fe_{0.9}Co_{0.1}OF$ nanorod and STEM-EDS elemental mapping of Fe, Co, O, and F. Inset of **a**: the crystal structure of $Fe_{0.9}Co_{0.1}OF$. The insets of **c**: corresponding SAED pattern. The insets of **d**: enlargement HRTEM image of the HRTEM image. The rectangle in **e** denotes the EDS mapping area in **f**. Scale bar in **b**–**e** is 400, 50, 5, and 50 nm, respectively

examined using X-ray diffraction (XRD), scanning electron microscopy (SEM), transmission electron microscopy (TEM) and high-resolution (HR) TEM, respectively. Figure 1a shows the XRD pattern of $Fe_{0.9}Co_{0.1}OF$, which has the same tetragonal rutile structure as FeOF with a slightly increased unit cell size (Supplementary Fig. 4). SEM (Fig. 1b) image reveals a uniform nanorod morphology of $Fe_{0.9}Co_{0.1}OF$ with a diameter of about 40–50 nm and length of 300–400 nm. Under the same synthesis conditions, the FeOF particle is much larger (Supplementary Fig. 5). However, by changing the solvent and the temperature, FeOF nanorods with the same size as $Fe_{0.9}Co_{0.1}OF$ was obtained (Supplementary Fig. 6). A representative TEM image (Fig. 1c) and corresponding selected area electron diffraction (SAED) pattern at inset show the growth direction as <112>. HRTEM image (Fig. 1d) in Fig. 1c further reveal the lattice spacing of 0.33 nm ((110) plane) for one-dimensional $Fe_{0.9}Co_{0.1}OF$ nanorods. The elemental distribution of nanorod was investigated with an energy-dispersive X-ray spectroscopy (EDS) technique in scanning TEM (STEM) mode. A high-angle annular dark-field (HAADF)-STEM image (Fig. 1e) and STEM-EDS elemental mappings (Fig. 1f) show that all four elements (Fe, Co, O, and F) are uniformly distributed throughout the nanorod, which is consistent with the cross-sectional STEM-EDS line profile of $Fe_{0.9}Co_{0.1}OF$ nanorod (Supplementary Fig. 7). EDS results also show that the atomic ratio of the O:F in $Fe_{0.9}Co_{0.1}OF$ and FeOF is almost the same and close to 1 (Supplementary Fig. 8).

**Electrochemical performance.** An unexpected high cycling stability is achieved for $Fe_{0.9}Co_{0.1}OF$ nanorods. Figure 2a displays a stable capacity of 350 mAh g$^{-1}$ at a high current of 500 mA g$^{-1}$ for 1000 cycles, while a higher capacity of 420 mAh g$^{-1}$ can be achieved at a low current of 70 mA g$^{-1}$ for over 330 cycles (Supplementary Fig. 9). The capacity's decay is about 0.005% at a high current of 500 mA g$^{-1}$, which is unprecedented for any conversion reaction materials and comparable to that of intercalation cathodes. The superior cycling stability is also evidenced by the almost identical discharge profiles at the 200th to the

curves at the 100th and the 3rd cycles (Fig. 2b). The charge/discharge profiles of $Fe_{0.9}Co_{0.1}OF$ cathodes at 70 mA g$^{-1}$ (Fig. 2b) are characterized by a declining discharge pseudo-plateau at 3.0 V followed by an extrusion reaction sloping line. In contrast, the $FeF_3$ has a similar initial capacity of ~400 mAh g$^{-1}$ at the same current of 70 mA g$^{-1}$ but it suffers a rapid capacity loss to 120 mAh g$^{-1}$ after only 25 cycles (Supplementary Fig. 9), which is in line with the reported performance[4]. The electrochemical performance of FeOF with different nanorod size is shown in Supplementary Fig. 10. Two FeOF with different sizes show a similar initial capacity of 450 mAh g$^{-1}$ that gradually reduced to 300 mAh g$^{-1}$ within 100 cycles at a current of 70 mA g$^{-1}$ (Supplementary Fig. 10).

$Fe_{0.9}Co_{0.1}OF$ nanorods show much higher rate capability than those of $FeF_3$ and FeOF (Fig. 2c, and Supplementary Fig. 11)[39,40]. $Fe_{0.9}Co_{0.1}OF$ delivers an enhanced capacity of ~440 mAh g$^{-1}$ at 80 mA g$^{-1}$. When the current is increased to 160, 320 and 640 mA g$^{-1}$, $Fe_{0.9}Co_{0.1}OF$ still retains a reversible capacity of ~400, ~380, and ~340 mAh g$^{-1}$ (Fig. 2c and Supplementary Fig. 11), respectively, whereas FeOF shows only 300, 250, and 200 mAh g$^{-1}$ at the same rates, and $FeF_3$ only possess a reversible capacity of ~60 mAh g$^{-1}$ at 320 mA g$^{-1}$. Figure 2d shows the Ragone plot of $FeF_3$, FeOF, and $Fe_{0.9}Co_{0.1}OF$ (based on active mass). At a lower specific power (a low discharge rate), FeOF or $Fe_{0.9}Co_{0.1}OF$ show a similar energy density of 1000 Wh kg$^{-1}$. However, $Fe_{0.9}Co_{0.1}OF$ shows the highest rate capability across the entire rate range: at 640 mA g$^{-1}$, the energy density of $Fe_{0.9}Co_{0.1}OF$ is twice that of FeOF, or six times higher than that of $FeF_3$, yielding the highest energy density ever reported for iron fluoride conversion reaction cathode materials at similar rates.

The electrochemical behavior of $Fe_{0.9}Co_{0.1}OF$ was characterized using cyclic voltammetry (CV) and galvanostatic intermittent titration technique (GITT). Figure 2e compares the CV scans of $Fe_{0.9}Co_{0.1}OF$, FeOF, and $FeF_3$. All three fluorides experiences a highly reversible intercalation reaction peaks at around 3.0 V. However, the followed conversion reaction peak at 1.6 V for $FeF_3$ shifted to a high potential of 1.9 V in FeOF and the peak current was significantly reduced. The conversion reaction

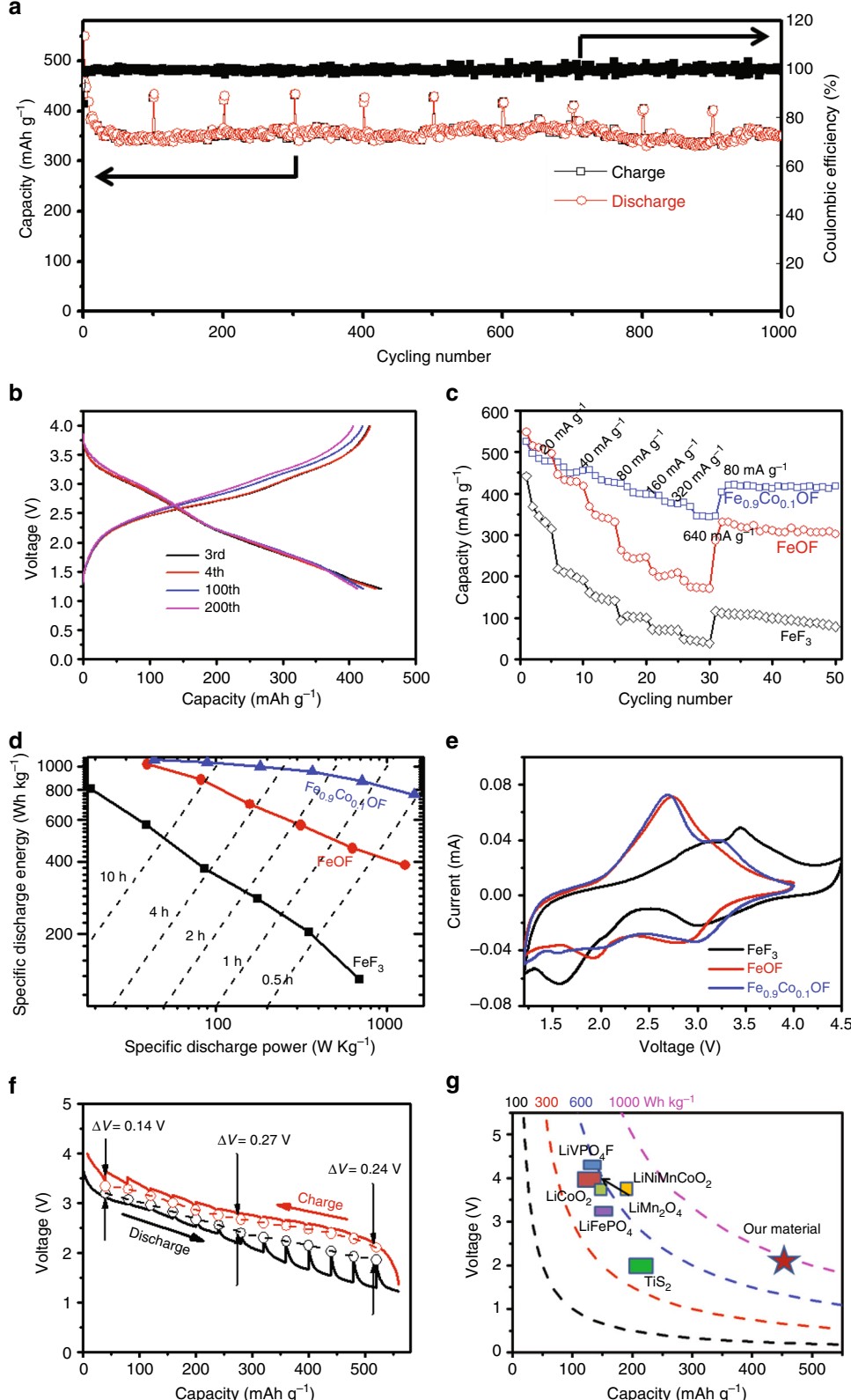

**Fig. 2** Electrochemical performance of the $Fe_{0.9}Co_{0.1}OF$ cathode. **a** Cycling performance for $Fe_{0.9}Co_{0.1}OF$ cathode with a current of 500 mA g$^{-1}$, every 100 cycles, the current changed to 70 mA g$^{-1}$ and cycled two cycles; **b** representative discharge/charge curves for $Fe_{0.9}Co_{0.1}OF$ at a current of 70 mA g$^{-1}$ in the potential range of 1.2–4.0 V; **c** rate capability for $Fe_{0.9}Co_{0.1}OF$, FeOF, and FeF$_3$ cathode materials, respectively; **d** the Ragone plot of FeF$_3$, FeOF, and $Fe_{0.9}Co_{0.1}OF$ (based on active mass); **e** CV profiles for the three materials (FeF$_3$, FeOF, and $Fe_{0.9}Co_{0.1}OF$) with a scanning rate of 0.1 mV s$^{-1}$. **f** Galvanostatic intermittent titration technique (GITT) measurement of $Fe_{0.9}Co_{0.1}OF$ (40 mA g$^{-1}$ for 1 h followed by a 20 h rest); **g** capacity, voltage, and energy density of $Fe_{0.9}Co_{0.1}OF$ (discharge energy density is calculated based on discharge current of 40 mA g$^{-1}$) as compared with the intercalation cathodes

peak completely disappeared in $Fe_{0.9}Co_{0.1}OF$, indicating that the conversion reaction was replaced by two extrusion reactions. During delithiation, two corresponding anodic peaks can be observed. The quasi-thermodynamic potential hysteresis of $Fe_{0.9}Co_{0.1}OF$ (Fig. 2f), FeOF, and $FeF_3$ (Supplementary Fig. 12) were measured using GITT. The hollow circles represent quasi-equilibrium potentials after fully relaxation at open circuit for 20 h, which approximates thermodynamic values. At 50% state of charge/discharge, this potential hysteresis of $Fe_{0.9}Co_{0.1}OF$ is only 0.27 V (Fig. 2f), which is substantially lower than 1.1 V for $FeF_3$ and 0.57 V for FeOF (Supplementary Fig. 12), respectively, and is even comparable to some intercalation-type cathodes[41]. The over-potential in $Fe_{0.9}Co_{0.1}OF$ (170 mV) is also significantly smaller than the reaction in FeOF (286 mV) and $FeF_3$ (634 mV) (Supplementary Fig. 13), and comparable to the lowest over-potential ever reported in mixed metal fluoride cathode ($Cu_yFe_{1-y}F_2$, 150 mV)[16]. The high potential hysteresis (>0.5 V) for conversion fluoride cathodes is attributed to the different conversion reaction pathways during lithiation and delithiation[6,7,20], which was considered as an intrinsic challenge and cannot be overcome through nanostructure design. We overcome this intrinsic issue by replacing the conversion reaction with two extrusion reactions through Co and O co-substitution in $FeF_3$, therefore the over-potential value of $Fe_{0.9}Co_{0.1}OF$ calculated from the middle points of the charge/discharge capacities is only <1/3 of the $FeF_3$ (Supplementary Fig. 14), resulting in an energy round-trip efficiency of >80% (Supplementary Fig. 15).

Non-lithiated $Fe_{0.9}Co_{0.1}OF$ cathode would require a lithiated anode in a full battery as its electrochemical couple, which is extremely difficult from perspective of practical production. For an ideal cathode material to be used as drop-in replacement by LIB industry, it is desirable to exist with built-in lithium source. Hence we pre-lithiated $Fe_{0.9}Co_{0.1}OF$ via ball milling with LiH (Supplementary Fig. 16). The charge/discharge curves of pre-lithiated $Fe_{0.9}Co_{0.1}OF$ (Supplementary Fig. 17) exhibit a higher over-potential than its non-lithiated counterpart in the first cycle. In the subsequent cycles, however, the voltage profiles of pre-lithiated $Fe_{0.9}Co_{0.1}OF$ (Supplementary Fig. 17) became almost identical to its unlithiated $Fe_{0.9}Co_{0.1}OF$ (Fig. 2b), with a much smaller voltage hysteresis than the pre-lithiated $FeF_3$ reported previously[20,40,42]. The cycling performance for the pre-lithiated $Fe_{0.9}Co_{0.1}OF$ is also similar to the non-lithiated one (Supplementary Fig. 18). The high capacity combined with a high discharge voltage leads to a discharge energy density of ~1000 Wh kg$^{-1}$ at current of 40 mA g$^{-1}$, which is higher than most intercalation cathode materials (Fig. 2g).

**PDF and ex situ TEM characterization**. The phase evolution of $Fe_{0.9}Co_{0.1}OF$ during (de)lithiation was analyzed using X-ray pair distribution function (PDF) and compared to that of FeOF. PDF analysis is a powerful tool to study local Fe ordering in FeOF and $Fe_{0.9}Co_{0.1}OF$[38,43,44]. Figure 3a displays the PDF patterns of three Fe-containing phases (rutile, rocksalt, and bcc Fe metal), and the corresponding atomic structures. Figure 3b shows the experimental PDF patterns of FeOF and $Fe_{0.9}Co_{0.1}OF$ at pristine state, charged state after the 1st cycle, and charged state after the 100th cycle. Apparently, all PDF patterns at charged states are similar to that at the pristine, indicating reversible rutile phase formation at charged states in both FeOF and $Fe_{0.9}Co_{0.1}OF$. In contrast, significant differences in PDF patterns arise at the discharged states (Fig. 3c). In these PDF patterns, the first peak at about 2.0 Å corresponds to the Fe–O and Fe–F correlation in the oxidized Fe phases (rutile and rocksalt), and the second peak at ~2.5 Å as indicated by the red arrows in Fig. 3c corresponds to the

nearest Fe–Fe pairs in metallic phase[38]. The PDF pattern of the discharged FeOF (1st cycle) can be fitted well using rocksalt and metal phases (Supplementary Fig. 19), which contains about 60.6% of rocksalt phase and 39.4% of Fe (Fig. 3d). The metallic Fe (39.4%) in FeOF is lower (50%) than that in oxygen-less $FeO_xF_{2-x}$ ($0.4 < x < 0.7$) materials[38]. In discharged $Fe_{0.9}Co_{0.1}OF$, the portion of metallic phase was further reduced to only about 13.1% while the rocksalt phase increases to 86.9% (Fig. 3d and Supplementary Fig. 19). After 100 cycles, the characteristic peak of metal in $Fe_{0.9}Co_{0.1}OF$ becomes a broadened shoulder (Fig. 3c). Fitting results indicate that the content of metallic Fe(Co) phase was significantly reduced to about 5% while the rocksalt phase dramatically increased to 95% (Fig. 3e). According to Wiaderek et al.[38], the lithiation of FeOF first goes through Li intercalation followed by phase changes from rutile to rocksalt with the extrusions of Fe and LiF, and then the rocksalt phase completely decomposes into $Li_2O$ and Fe. The PDF results suggest that the reaction natures of $Fe_{0.9}Co_{0.1}OF$ and FeOF are different: for $Fe_{0.9}Co_{0.1}OF$, conversion reaction is suppressed and the reversibility of extrusion reaction II is enhanced by suppressing the formation metallic phase through co-substitution. It indicates that the main reaction in $Fe_{0.9}Co_{0.1}OF$ is an intercalation-extrusion reaction between rutile and rocksalt phase (Eq. 4). The mechanism for structure enhanced electrochemical performance will be discussed later.

The morphology and phase upon cycling were investigated using HRTEM, STEM, and SAED (Supplementary Fig. 20 and Fig. 3f–h). When $Fe_{0.9}Co_{0.1}OF$ was discharged to 1.2 V, bright contrast of small nanoparticles from STEM imaging was observed and its SAED pattern indicates that the main phase of discharged $Fe_{0.9}Co_{0.1}OF$ is rocksalt (Fig. 3f). After being charged back to 4 V, rocksalt phase changed back to the rutile phase (Fig. 3g). In the discharged state after 100 cycles (Fig. 3h), the rocksalt phase still dominates $Fe_{0.9}Co_{0.1}OF$, which is consistent to the PDF results and previous reports[38,45]. In these SAEDs, we observed the spotty-like diffraction patterns indicating large sizes of rutile/rocksalt grains, whereas in HAADF-STEM images, the small bright nanoparticles and its corresponding broaden rings in diffraction patterns[33,46] are attributed to the metallic nanoparticles. Considering HRTEM images and their fast Fourier transform (Supplementary Fig. 20), we conclude that the rocksalt phase accounts for the main part of discharged $Fe_{0.9}Co_{0.1}OF$ with small amount of metallic Fe(Co). In addition, the Co-doping in FeOF reduces the size of metallic nanoparticles in discharged $Fe_{0.9}Co_{0.1}OF$. Figure 3i compares the size distributions of nanoparticles formed in FeOF and $Fe_{0.9}Co_{0.1}OF$ after second discharging to 1.2 V, where the Fe/Co particle size from $Fe_{0.9}Co_{0.1}OF$ are between 0.5 and 3.5 nm with an average size of 1.8 nm, while the Fe particles in dicharged FeOF widely distributes between 2 and 5.5 nm with average size of 3 nm (Supplementary Fig. 21). In comparison, those decomposed from $FeF_3$ are around 5 nm[13]. Figure 3j show that the rocksalt phase ($Li_xFeO_{2-y}$), Fe metal, and LiF have similar lattice constants. The cubic-on-cubic coherence of their lattice can help to form a composite to maintain the overall shape of nanorods, thus maintaining morphology and structure integration during cycling.

**In situ TEM investigation of structural evolution**. The structural evolution of $Fe_{0.9}Co_{0.1}OF$ during cycling was further investigated by in situ TEM technique with an experimental setup schematically illustrated in Fig. 4a. The time-lapse images and movie for lithiation are shown in Fig. 4b, and Supplementary Movie 1, and those for delithiation are shown in Supplementary Fig. 22 and Supplementary Movie 2, respectively. A clear contrast

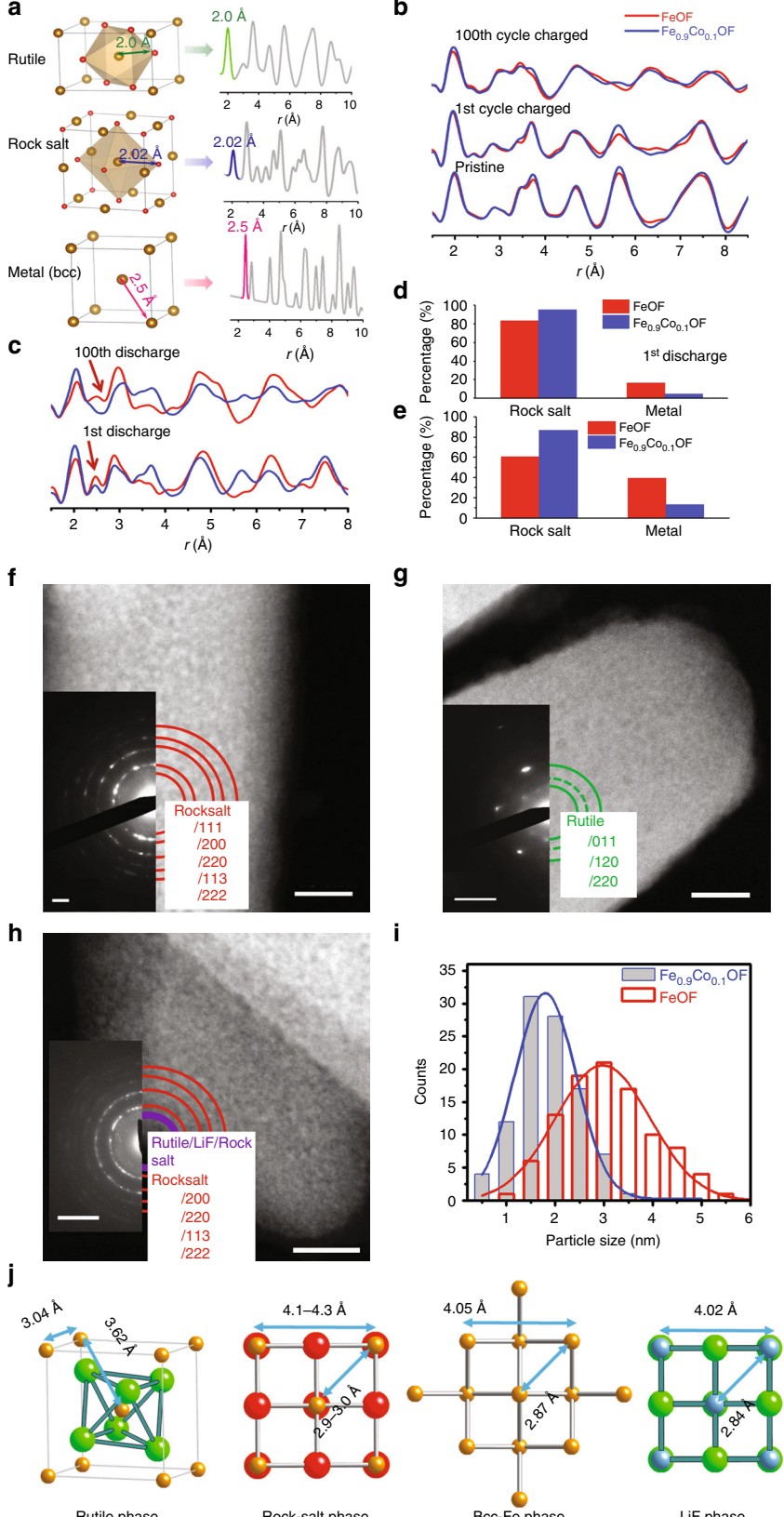

**Fig. 3** Microstructure evolution during cycling for the $Fe_{0.9}Co_{0.1}OF$ cathode material. **a** Illustration of the correspondence between PDF peaks and specific atomic pairs. **b**, **c** PDF patterns of $Fe_{0.9}Co_{0.1}OF$ and FeOF for different charge and discharge states. **d**, **e** The concentration of rocksalt phase, and the metal phase by fitting PDF results after different discharge states. **f**–**h** HRTEM images of $Fe_{0.9}Co_{0.1}OF$ electrode after discharged to 1.2 V, charged to 4 V, and after 100 cycles discharged, respectively. **i** The histograms of the metal particle size distribution for $Fe_{0.9}Co_{0.1}OF$ and FeOF electrodes after discharged to 1.2 V. Insets in **f**–**h** are the corresponding SAED patterns. **j** Illustration of the crystal structures of the four structures during cycling. Scale bar in **f**–**h** is 10 nm. Scale bar in the inset of **f**–**h** is 2, 5, and 5 nm$^{-1}$, respectively

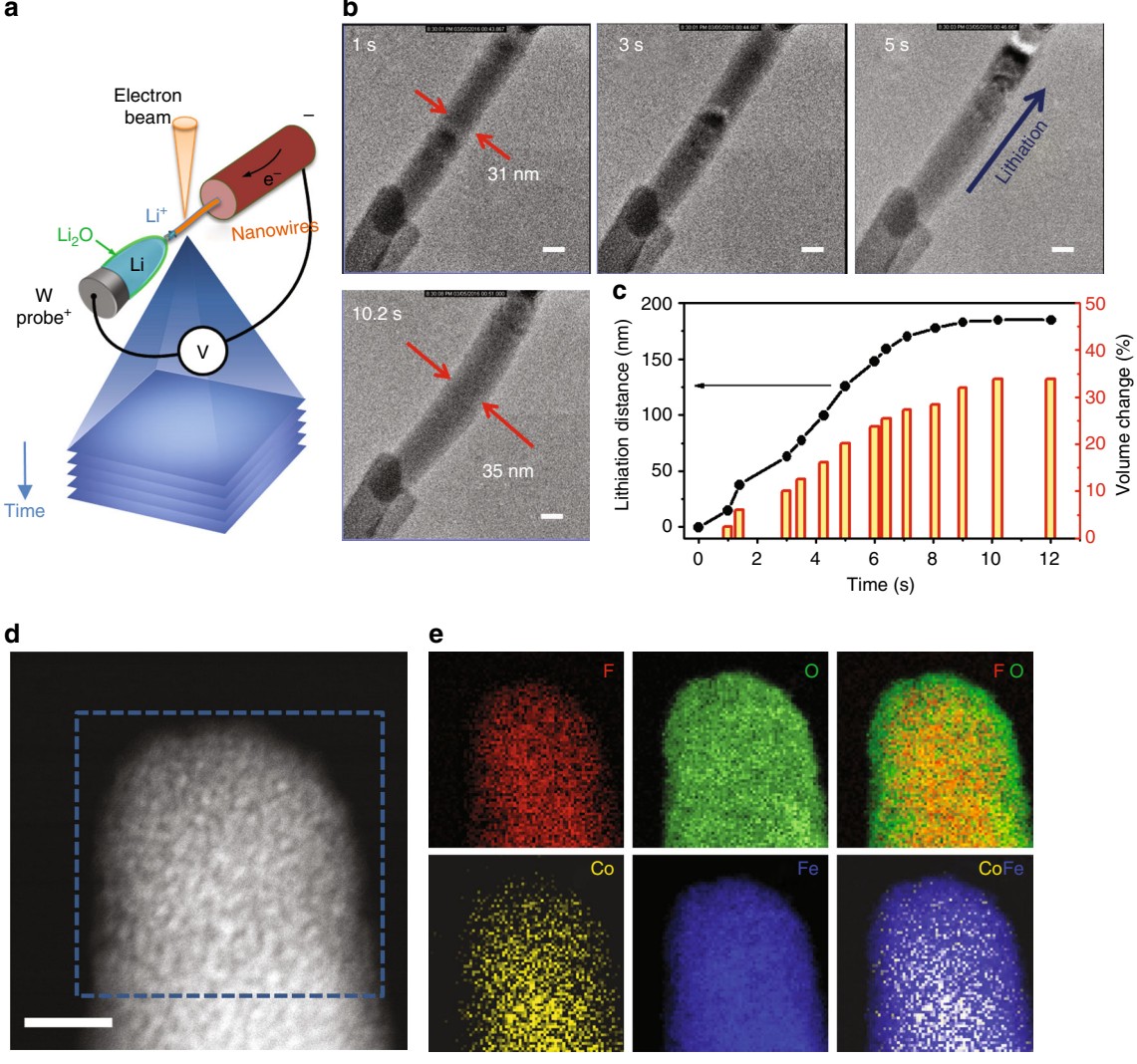

**Fig. 4** Structure conversion of $Fe_{0.9}Co_{0.1}OF$ nanorod monitored in real time and the elemental composition distribution. **a** Schematic illustration of the electrochemical setup for in situ TEM test. **b** In situ HRTEM image of $Fe_{0.9}Co_{0.1}OF$ during dynamic lithiation at 1.2 V for different time. **c** The lithiation distance and the volume change as function of time. **d** ADF-STEM image and **e** corresponding Fe, Co, O, and F elemental distributions after the first lithiation obtained by STEM-EELS technique. Scale bar in **b** and **d** is 20 and 10 nm, respectively

change was observed during Li insertion reaction[46,47], and the reaction front moves rapidly across the nanorod (from the lower left to upper right). Compared with the particle-to-particle diffusion in $FeF_3$[47], $Fe_{0.9}Co_{0.1}OF$ exhibits a much faster reaction rate without any obvious variations on surface and in bulk. Interestingly, the lithiation of a 200 nm $Fe_{0.9}Co_{0.1}OF$ nanorod occurs within only ~10 s. In comparison, for $FeF_3$ nanoparticles with much smaller size of ~10 nm, it takes at least a few minutes for full lithiation[47]. Figure 4c compares the lithiation distance and volume change vs. time. It agrees with the electrochemical performances shown in Fig. 2c, d, and indicates that the reaction kinetics after Co/O co-substitution is largely improved[12,48]. During lithiation of $Fe_{0.9}Co_{0.1}OF$, the nanorod volume expands by ~35% with a little bend due to the stress, but no obvious cracks are observed.

The STEM-EDS results have confirmed that Fe and Co elements are homogeneously dispersed in $Fe_{0.9}Co_{0.1}OF$ nanorods at both pristine (Fig. 1, Supplementary Fig. 7) and charged states (Supplementary Fig. 23). In addition, the chemical distributions of $Fe_{0.9}Co_{0.1}OF$ upon cycling have been investigated using STEM-electron energy-loss spectroscopy (EELS) mapping and line-scan (Fig. 4d, e and Supplementary Fig. 24). It is found that the surface

of lithiated $Fe_{0.9}Co_{0.1}OF$ is covered with 1.5~2 nm oxygen-rich surface layer (Fig. 4e), which is still observed even after 100 cycles (Supplementary Fig. 24c). In Supplementary Fig. 24c, at the surface (position of 8 nm), the O content is much higher than F content. However, no such layer was observed in FeOF. Instead, nano-sized domains with size of 10–20 nm were observed, where significant element segregations can be detected (Supplementary Fig. 25). We believe that the surface O-rich rocksalt layer has been effectively stabilized by Co-doping, which consequently inhibits the reaction of electrolyte with the nascent metal nanoparticles and the subsequent dissolution of transition metals into electrolytes[33]. This conclusion is evidenced by the elemental analysis using inductively coupled plasma atomic emission spectroscopy (Supplementary Fig. 26), where a much lower transition metal concentration is detected in the electrolytes for $Fe_{0.9}Co_{0.1}OF$ than that for FeOF after prolonged cycling. It should be noted that this O-rich oxide layer with thickness of ~2 nm will not retard the Li insertion in later cycles according to DFT energy barrier calculations (Supplementary Fig. 27 and Supplementary Note 1). The diffusion barrier in $LiFeO_2$ rocksalt phase is calculated to be 0.29 eV, which is similar to that in $LiFePO_4$ cathode materials[49].

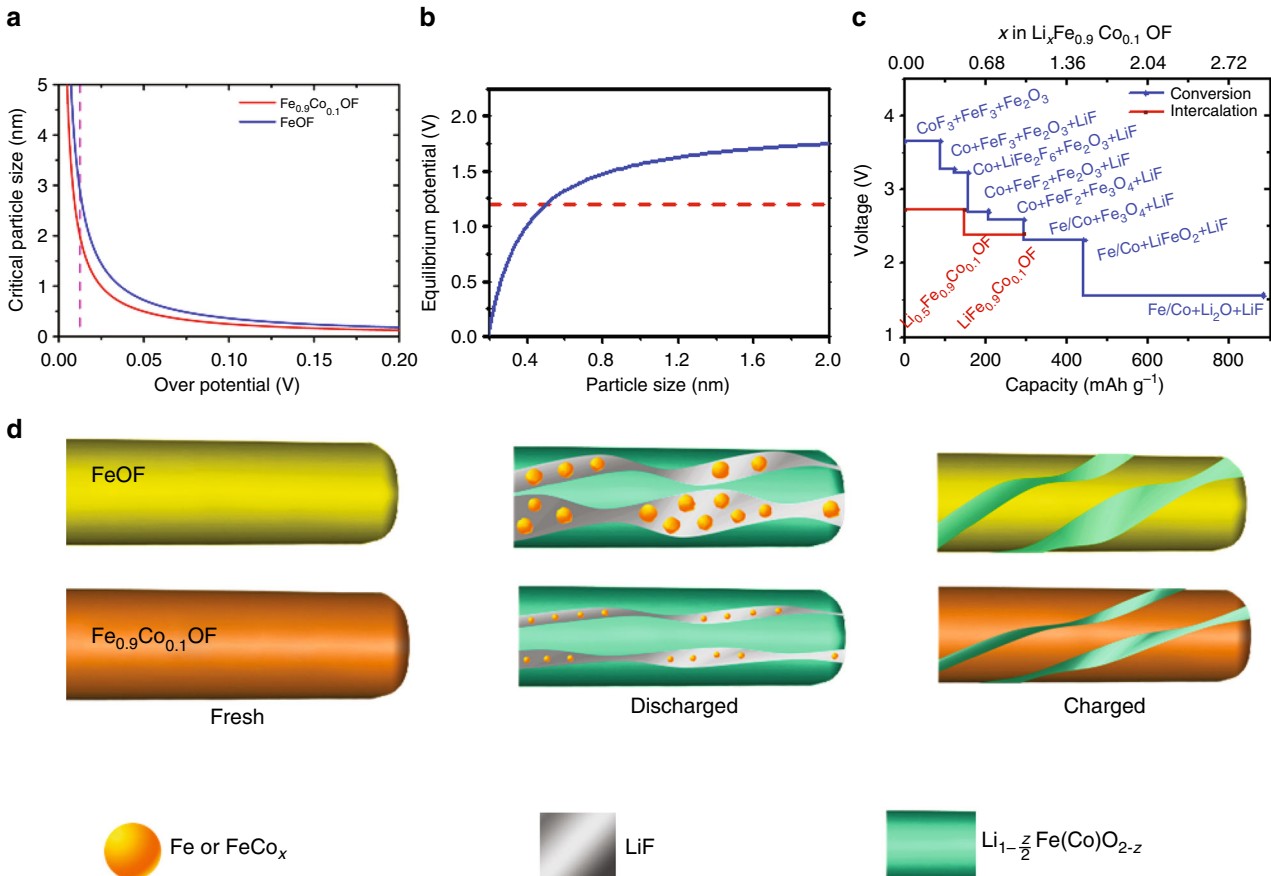

**Fig. 5** Reaction mechanism for the $Fe_{0.9}Co_{0.1}OF$ cathode. **a** The relationship between the overpotentials and the metal critical particle sizes during lithiation for FeOF and $Fe_{0.9}Co_{0.1}OF$. **b** The relationship between the in situ-formed metal particle size with the equilibrium potential of the conversion reaction. **c** Calculation reaction curves of the $Fe_{0.9}Co_{0.1}OF$ materials from the DFT intercalation of Li into $Fe_{0.9}Co_{0.1}OF$, and the conversion path using the equilibrium phases (assuming the size of metal particles is 1 nm). **d** Illustrations of structural evolution for FeOF and $Fe_{0.9}Co_{0.1}OF$ during lithiation

**Reaction mechanism of $Fe_{0.9}Co_{0.1}OF$.** From the structural characterizations using PDF and TEM, we have observed the following changes after doping Co into FeOF: (1) after the first fully lithiation, the amount of metal reduced from 39.4 to 13% while the amount of rocksalt phase significantly increased from 60.6 to ~86.9%. After 100 cycles, the amount of metallic phase in lithiated $Fe_{0.9}Co_{0.1}OF$ was further reduced to ~5%. (2) The particle size of $FeCo_x$ in discharged state reduced from 3.0 to 1.8 nm and the $FeCo_x$ particle size was further reduced during charge/discharge cycles and almost not detectable after 100 cycles. (3) $Fe_{0.9}Co_{0.1}OF$ possesses a stable oxygen-rich surface layer during long-term cycling.

First, it should be pointed out that similar to the $FeF_3$, the surface energy barrier has minor impact on the Li diffusion in the $Fe_{0.9}Co_{0.1}OF$ system according to the DFT calculation results (Supplementary Fig. 28, Supplementary Note 2, Supplementary Table 1, and Supplementary Note 3). The surface diffusion barrier (0.88 eV) is lower than the bulk diffusion barrier (1.36 eV) in $Fe_{0.9}Co_{0.1}OF$. To decipher the mechanism for reducing the Fe particle size from Co substitution, we calculated the nucleation rate of metallic phase from interface energies during the lithiation (Supplementary Fig. 29), and the formation energy of Fe and $FeCo_x$ particles using first-principles calculations. These calculations indicate that the formation of $FeCo_x$ nanoparticles becomes much easier in $Fe_{0.9}Co_{0.1}OF$, significantly reducing the critical nucleation size of $FeCo_x$ particles in $Fe_{0.9}Co_{0.1}OF$ compared to Fe particles in FeOF (Fig. 5a). The critical particle sizes of $FeCo_x$ and

Fe at an over-potential of 0.043 V are 1.8 and 2.9 nm, respectively (indicated by the dashed line in Fig. 5a), corresponding to a ratio of 3.4 for the number of particles in $Fe_{0.9}Co_{0.1}OF$ over FeOF (Eq. 13), which is in consistence with the observed $FeCo_x$ and Fe sizes (Fig. 3i). The reduced Fe particle size from 3.0 nm in FeOF to 1.8 nm in $Fe_{0.9}Co_{0.1}OF$ also reduces the thermodynamic potential for conversion reaction from 1.812 to 1.729 V (Fig. 5c). This conversion potential will be further reduced with cycling due to continuous reduction of $FeCo_x$ metal nanoparticle size and amount during cycles (Fig. 5b), thus conversion reaction in $Fe_{0.9}Co_{0.1}OF$ is completely suppressed. The reduction of content and size of $FeCo_x$ metal nanoparticle in $Fe_{0.9}Co_{0.1}OF$ is also accompanied by formation of non-stoichiometric rocksalt phase, which significantly enhance the reversibility of extrusion II (Eq. 4). This is confirmed by disappearance of conversion peaks in CV for $Fe_{0.9}Co_{0.1}OF$ (Fig. 2e) and intercalation and extrusion sloping curves in galvanostatic charge/discharge (Fig. 2b).

DFT calculations (Fig. 5c, and Supplementary Fig. 30) show the initial lithiation process an intercalation reaction (red line in Fig. 5c and Supplementary Fig. 30), and then the LiF and small amount of Fe/Co extrude out of the $LiFeO_2$ rocksalt phase, which is consistent to the PDF and TEM investigations (Fig. 3). Theoretically, all levels of lithiation should be "conversion phases", as shown the blue line in Fig. 5c, and Supplementary Fig. 30. However, given that the small energy difference between the intercalation and the conversion reaction, and the large kinetic barriers to the conversion reactions[34], it is reasonable that

the material intercalate lithium before it converts (red line), which is indeed in consistence with the experiment. Considering the PDF analysis of FeOF by Wiaderek et al.[38], we propose the lithiation of $Fe_{0.9}Co_{0.1}OF$ will go through a similar process, involving intercalation and two steps of extrusion reactions (Eq. 4).

On the basis of DFT calculations, ex situ/in situ TEM, and PDF analyses, the lithiation/delithiation of $Fe_{0.9}Co_{0.1}OF$ in the first charge-discharge cycle can be described by Eq. 4. The fully lithiated $Fe_{0.9}Co_{0.1}OF$ consist of 87% of non-stoichiometric $Li_{0.576}Fe(Co)O_{1.15}$ and 13% of Fe(Co) as quantitatively demonstrated by PDF. The extrusion of LiF from the lithiated materials, which is similar to the lithiation reaction in layered material of $Cu_{2.33}V_4O_{11}$[50], has fast reaction kinetics without metal accumulation due to low content of Fe(Co). After 100 cycles, the extrusion reaction II become more reversible and fast due to continues reduction of Fe(Co) content (~5%) and size. However, due to the highly defected rocksalt phase, similar capacities were achieved in the cycling. Meanwhile, the lattice coherence of discharged compounds of LiF, rocksalt phase, and metallic phase reduces the interfacial energy and help to improve the reversibility of the compound[34].

A schematic illustration of comparing the (de)lithiation mechanisms of FeOF and $Fe_{0.9}Co_{0.1}OF$ is shown in Fig. 5d. When FeOF is fully discharged to a potential of 1.2 V, the FeOF experiences four reactions (Li intercalation, extrusion I, extrusion II, and partial conversion reaction) (Eq. 3). The existence of conversion reaction in the lithiated FeOF after being discharged to 1.2 V still limited the cycling stability due to the high irreversibility of conversion reaction. In contrast, the conversion reaction in $Fe_{0.9}Co_{0.1}OF$ is completely inhibited and the extrusion reaction II becomes more reversible with charge/discharge cycles. As shown in Supplementary Fig. 31 and Supplementary Note 4 and Note 5, the smaller particle size ensures almost half of the atoms to be located at the interface layer for $Fe_{0.9}Co_{0.1}OF$ after lithiation, thereby facilities the extrusion reaction and limited the diffusion distance for different ions. This pronounced reduction in particle sizes and stable coherent structures between LiF, metallic phase, and rock phase (Fig. 3d), together with the stabilized O-rich rocksalt surface layer (Fig. 4e, Supplementary Fig. 24) ensure a highly homogeneous distribution of various reaction ingredients, which was proposed as a key rationale behind the unprecedented reversibility[51]. During charge (delithiation), LiF and very small amount of $FeCo_x$ as well as rocksalt phase will convert back to rutile phase. It is worth mentioning that such reaction mechanism does not always apply for all of the fluoride systems. For example, Wang et al.[16] showed that there was no intercalation reaction in the $Cu_yFe_{1-y}F_2$ materials, suggesting that the nature of reaction mechanism depends on transition metal chemistries. Such dependence in fact provides possibility of tuning the reaction route through elemental doping and tailoring the synthesis route, which is exactly the motivation of this work.

## Discussion

In summary, we successfully improved the cycling stability and reaction kinetics of $FeF_3$ by a Co and O co-substitution strategy. We found that the co-substituted $Fe_{0.9}Co_{0.1}OF$ can achieve an energy density as high as 1000 Wh $kg^{-1}$ (420 mAh $g^{-1}$) at 70 mA $g^{-1}$ for 330 cycles, or 350 mAh $g^{-1}$ for 1000 cycles at a high rate of 500 mA $g^{-1}$. In addition, Co and O substitution of $FeF_3$ also significantly reduced the potential hysteresis to 0.27 V. In-depth analysis using analytical/in situ TEM techniques, PDF analysis, and first-principles calculations have revealed that the co-substituted anion (O) and cation (Co) in $Fe_{0.9}Co_{0.1}OF$ reduced

the particles size and abundances of metallic Fe(Co), thus suppressed less-reversible conversion reaction and further enhanced the reversibility of extrusion reactions. The highly reversible intercalation-extrusion reaction pathway of $Fe_{0.9}Co_{0.1}OF$ enhances the cycling stability and reaction kinetics. Therefore, the $Fe_{0.9}Co_{0.1}OF$ nanorods achieved exceptional cycling performance with the small hysteresis, highest round-trip energy efficiency, and improved kinetics, which can meet the critical requirements of applications for next-generation cathode materials. The effective co-substitution strategy could be applied to resolve similar reversibility issues encountered by other conversion reaction materials, such as oxides and sulfides.

## Methods

**Material synthesis.** $Fe_{0.9}Co_{0.1}OF$ and FeOF were synthesized via solvothermal method. Typically, $FeF_3 \cdot 3H_2O$ and $CoF_3$ were ball milled at a molar ratio of 9:1 for 30 min to reduce the particle size and get a homogeneous reactant mixture. Then, 170 mg of such mixture was dispersed in 1-propanol (75 ml) and stirred vigorously at room temperature for half an hour. The resulting suspension was transferred into a Teflon-lined 100 ml stainless-steel autoclave reactor, and subsequently sealed and heated at 210 °C for 24 h in an oven. For comparison, FeOF was also synthesized by the same procedure using pure $FeF_3 \cdot 3H_2O$ as the starting material dispersed in butyl-alcohol. $FeF_3$ was prepared by ball milling the as-purchased $FeF_3$ for 6 h. The pre-lithiation of $Fe_{0.9}Co_{0.1}OF$ is synthesized by ball milling $Fe_{0.9}Co_{0.1}OF$ and LiH at a molar ratio of 1:2 for 10 h.

**Material characterizations.** Microstructural analyses were performed by using SEM (Hitachi SU-70) and STEM (JEOL 2100 field emission gun TEM, operated at accelerating voltage of 200 kV), respectively. XRD pattern was recorded by Bruker Smart1000 (Bruker AXS Inc., USA) using Cu Kα radiation.

PDF experiments were carried out at X-ray powder diffraction (XPD) beamline (ID28) at the National Synchrotron Light Source II, Brookhaven National Laboratory, USA, with a photon wavelength of 0.185794 Å. A large-area amorphous-silicon-based detector was used to collect data to high values of momentum transfer ($Q_{max} = 24$ Å$^{-1}$). The raw images were integrated using the software FIT2d[52]. PDFgetX3[53] was used to correct the data for background contributions, Compton scattering, and detector effects, and to Fourier transform the data to generate $G_{(r)}$, the PDF.

$$G_{(r)} = 4\pi r[\rho_{(r)} - \rho_0] = \frac{2}{\pi} \int_0^\infty Q[S_{(Q)} - 1] \sin(Qr)dQ \qquad (5)$$

Here $\rho_{(r)}$ is the microscopic pair density, $\rho_0$ is the average number density, and $Q$ is magnitude of the scattering vector. For elastic scattering $Q = 4\pi\sin(\theta)/\lambda$ with $2\theta$ being the scattering angle and $\lambda$ the wavelength of the radiation used. $S_{(Q)}$ is the total scattering function. The intensity and position of peaks corresponding to the Fe–Fe/Co bond in metallic phase (~2.5 Å) were fitted by using Gaussian functions within fityk[54]. Structure models were refined against the PDF data within PDFgui[55].

**In situ TEM.** A nanofactory scanning tunneling microscopy-TEM holder was used in the experiment. The holder is equipped with three-dimensional piezo-manipulator and biasing capability. $Fe_{0.9}Co_{0.1}OF$ nanorods were attached on a tungsten probe using conducting epoxy and mounted on one side of the holder. On the other side, we mounted another tungsten rod after transferring a small piece of Li on its tip. The $Fe_{0.9}Co_{0.1}OF$ nanorods and Li metal were then brought into contact inside the TEM. By applying voltage on the working electrode vs. the counter electrode (Li), lithiation and delithiation processes were recorded. The experiment was performed using a JEOL 2010F TEM operating at accelerating voltage of 200 kV. The HAADF-STEM imaging, STEM-EELS, and STEM-EDX were performed with an aberration-corrected Hitachi HD2700C STEM at 200 kV in Brookhaven National Lab. The conversion angle and collection angles for STEM imaging are 22 and 64–341 mrad, respectively. The collection angle of EELS are from 20 to 26 mrad depending on the needs.

**First-principles calculation methods.** All DFT calculations were performed using the Vienna Ab initio Simulation Package[56] within the projector augmented-wave approach[57], and the Perdew–Burke–Ernzerhof generalized gradient approximation (GGA) functionals[58] was used. In structures containing Fe (Co) and O or F, a value of $U = 4.0$ eV (3.32 eV) was used for Fe (Co), as pervious DFT studies. Spin-polarized total energy calculations and structure relaxations were performed. Fe and Co atoms were initialized in high-spin ferromagnetic ordering for simplicity. And the parameters of DFT calculations, such as the plane-wave energy cutoff and $k$-point density, were consistent with the parameters used for the Materials Project (MP). The voltage plateaus are obtained using the calculated DFT energies of all relevant compounds in the Li-Fe-O-F space from the MP[59]. The structural matching results were visualized using VESTA[60].

**Energy of metal nanoparticles**. For the reason that GGA does not capture well the energy difference between the localized $d$-states for ionic iron and cobalt in oxide or fluoride and the delocalized states in metallic iron and cobalt. To solve this problem, as proposed by Doe, the experimentally measured reaction enthalpy obtained for $MF_2 + 2Li = M + 2LiF$ (M = Fe and Co) and the energies for Li, $FeF_2$, and LiF are used to determine the energies of metallic Fe and Co[25]. This approach yields cohesive energies of −4.2811 and 4.3974 eV for bulk, metallic Fe and Co, respectively, corresponding to those obtained from experiment (i.e., −4.28 and −4.39 eV).

To evaluate the nanosize effect, the Wulff shapes of Fe and Co are constructed to give the crystal shape under equilibrium conditions. The Wulff shape is constructed by based on the elemental surface energy data form the database developed by Shyue Ping Ong group[61]. The weighted surface energy $\bar{\gamma}$ using this fraction is given by the following equation:

$$\bar{\gamma} = \frac{\sum_{\{hkl\}} \gamma_{hkl} A_{hkl}}{\sum A_{hkl}} \quad (6)$$

where $\gamma_{hkl}$ is the surface energy obtained from the database for a unique facet existing in the Wulff shape and $A_{hkl}$ is the total area of all facets in the {hkl} family in the Wulff shape. Pymatgen is used to generate slabs and calculate weighted surface energy[61].

The energy of metal nanoparticle can be calculated by the following equation:

$$E(\text{nano}) = E^{\text{Fit}}(\text{bulk}) - \bar{\gamma} \frac{3V_M}{rN_A} \quad (7)$$

**Over-potential and critical nucleation particle radius**. The particle size was calculated by using the same approach as previous reported[62]. For lithiation of FeOF and $Fe_{0.9}Co_{0.1}OF$ reaction below, over-potential is the driving force for nucleation. As shown in Figs. 2c and 3a, at a low discharge/charge rate, similar capacity (~480 mAh $g^{-1}$) can be obtained for FeOF and $Fe_{0.9}Co_{0.1}OF$, this capacity corresponds to 1.63 mol Li per 1 mol active materials (FeOF or $Fe_{0.9}Co_{0.1}OF$), with a decent difference of the metal particle size and the metal contents.

$$1.63\,Li + FeOF \rightarrow 0.39\,Fe + LiF + Li_{0.63}Fe_{0.61}O \quad (8)$$

$$1.63\,Li + Fe_{0.9}Co_{0.1}OF \rightarrow 0.13\,Fe_{0.9}Co_{0.1} \\ + LiF + Li_{0.63}M_{0.87}O \quad (9)$$

By assuming that the particle has a sphere shape, the formation energy $\Delta G$ of a particle can be written as a function of over-potential $\Delta\phi$ and particle size $r$.

$$\Delta G(\Delta\phi, r) = \frac{4}{3}\pi r^3 \cdot \frac{1}{V_M} \cdot \frac{1.63}{x} \cdot \Delta\phi \cdot N_A - \gamma \cdot 4\pi r^2 \quad (10)$$

where $V_M$ is molar volume of Fe or $Fe_{0.9}Co_{0.1}$ particle. $\gamma$ is the interface energy. $x$ is the coefficient of Fe ($x = 0.39$) or $Fe_{0.9}Co_{0.1}$ ($x = 0.13$) in reaction Eqs. (9) and (10), respectively. $N_A$ is the Avogadro constant. Specific $x$ values are derived based on PDF studies. As mentioned in the manuscript, the PDF pattern of discharged FeOF can be well fitted by a mixture of rocksalt and metal phases, directly yielding the phase fraction of metal phase (for FeOF, $x = 0.39$). For the Co-substituted one, the estimation of phase fraction was made based on the intensity of the characteristic peak of metal at around 2.5 Å (for $Fe_{0.9}Co_{0.1}OF$, $x = 0.13$).

The critical nucleus radius is calculated by

$$\frac{\partial G(\phi, r)}{\partial r} = 0 \quad (11)$$

Therefore we can get the critical nucleus radius $r^*$

$$r^*(\phi) = \frac{2\gamma \cdot x \cdot V_M}{1.63 \cdot \phi \cdot N_A} \quad (12)$$

The relationship between critical particle size and over-potential is shown in Fig. 4i.

Assuming that the same amount of metal particle is converted, then the ratio of nucleation number under same over-potential can be calculated as

$$\frac{N_{Fe_{0.9}Co_{0.1}}(\Delta\phi)}{N_{Fe}(\Delta\phi)} = \left(\frac{1/r^*_{Fe_{0.9}Co_{0.1}}(\Delta\phi)}{1/r^*_{Fe}(\Delta\phi)}\right)^3 \quad (13)$$

**Electrochemical tests**. The electrochemical tests were performed using a coin-type half cell (CR 2032). Metallic lithium was used as the negative electrode. To prepare

working electrode, the as-synthesized $Fe_{0.9}Co_{0.1}OF$ (or FeOF, $FeF_3$), carbon black, and polyvinylidene fluoride with mass ratio 70:15:15 were mixed into a homogeneous slurry in NMP with pestle and mortar. The slurry mixture was coated onto Al foil and then dried at 100 °C for 12 h under air atmosphere. For the pre-lithiated $Fe_{0.9}Co_{0.1}OF$ materials, all of the preparing procedures for the electrode were protected by Ar to avoid the possible oxidation of in situ-formed Fe nanoparticles. The electrolyte consists of 1.0 M $LiPF_6$ dissolved in mixture of fluoroethylene carbonate, 2,2,2-trifluoroethyl methyl carbonate, and 1,1,2,2-tetrafluoroethyl-2′,2′,3′,3′-tetrafluoropropyl-ether (HFE) at the weight ratio of 20:60:20. The cells were assembled with a polypropylene microporous film (Celgard 3501) as the separator. Electrochemical performance was tested using Arbin battery test station (BT2000, Arbin Instruments, USA). Capacity was calculated on the basis of the mass of active species ($Fe_{0.9}Co_{0.1}OF$, FeOF, or $FeF_3$). The energy density of the cathode was machine generated on the Arbin Instruments. Cyclic voltammogram scanned at 0.1 mV $s^{-1}$ between 1.2 and 4.5 V was recorded using a CHI 600E electrochemical workstation (CH Instruments Inc., USA).

**Data availability**. All relevant data are available from the authors on request.

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

## Acknowledgements

X.F. and C.W. acknowledge the financial support from Army Research Lab under Award Number W911NF1420031. Daikin America provided the high purity fluorinated solvents. The PDF data were collected at XPD beamline (28ID-2) of NSLSII, and the electron microscopy analysis was carried out at the Center for Functional Nanomaterials, Brookhaven National Laboratory (BNL), which is supported by the DOE, Office of Basic Energy Sciences, under contract DE-SC0012704. The PDF studies at BNL were supported by the Assistant Secretary for Energy Efficiency and Renewable Energy, Office of Vehicle Technologies of the U.S. Department of Energy through the Advanced Battery Materials Research (BMR) Program, including Battery500 Consortium under contract DE-SC0012704. C.W. and K.X. also acknowledge the support of EERE of USDOE through Battery500 Consortium Seeding project under contract DE-EE0008200. The authors acknowledge the University of Maryland supercomputing resources (http://hpcc.umd.edu) made available for conducting DFT computations in this paper. We also appreciate Dr. Qingping Meng at BNL, and Dr. Nancy J. Dudney at Oak Ridge National Laboratory (ORNL) for the constructive discussions.

## Author contributions

X.F. made the samples. E.H. and X.-Q.Y. performed the PDF experiments. X.J., Y.Z., and Y.M. conducted the calculations. F.H., S.H., J.L., S.B., Z.M., T.G., S.-C.L., and J.B. designed the experiments. K.X. screened and prepared the electrolytes. X.F., E.H., D.S.,

and C.W. analyzed the experiments. X.F. wrote the draft manuscript. E.H., D.S., K.X., and C.W. revised the manuscript.

## Additional information

**Competing interests:** The authors declare no competing interests.

