## [Peer Review File · Nature Communications]

Reviewers' comments:

Reviewer #1 (Remarks to the Author):

Manuscript NCOMMS-17-30638

The paper by Xiulin Fan et al entitled "High Energy-Density and Reversibility of Iron Fluoride Cathode Enabled Via an Intercalation-Extrusion Reaction" is a well written paper dealing with synthesis, characterization and electrochemical performance of Fe_{0.9}Co_{0.1}OF based cathodes. This paper follows in the steps of past developments on iron fluorides cathodes from FeF₂ to FeF₃ and FeOF which show continuing improvements in cycle life. In this paper, partial substitution of Fe by Co gives rise to further improvements in cycling performance and voltage hysteresis of iron fluoride cathodes. The results reported in this paper are excellent with detailed TEM and PDF characterization techniques and should be of interest to scientists in the energy storage field. This paper can be published in the present form with minor corrections.

A few general comments:

1. Mixed metal fluorides of composition Cu_yFe_{1-y}F₂ were previously investigated by Wang et al (doi:10.1038/ncomms7668) (Ref 14 in the present paper) where they show that with Cu substitution, they can achieved very low hysteresis (over potential) of less than 150 meV. Reference to this paper should be mentioned in the section on over potential starting with line 192 as this work is complementary to the Co substitution presented in this study which reports a decrease in voltage hysteresis of Fe_{0.9}Co_{0.1}OF as compared to FeOF.
2. Too many sub-figures are placed within single figures making them too small and totally unreadable. My recommendation is to increase the number of figures and enlarge their sizes in a way to make the lettering size as recommended by the publisher. This is especially true for Figure 2.
3. In Figure caption 4, please specify if the elemental distribution is obtained by STEM-EELS or EDX techniques.

Reviewer #2 (Remarks to the Author):

This is a very interesting paper in which the authors have done a significant number of experiments to evaluate the effect of adding Co to FeOF in order to change the deleterious hysteresis between charge and discharge. By removing the conversion reactions, which form Fe crystals in the conversion fluorides, the hysteresis appears to be significantly reduced. The work provides a new route to a higher capacity cathode and improves the previously considered Fe-fluorides and oxy-fluorides. Their compositional modifications may open the field to other additives that may also show further improvements. The paper needs some editing in wording, and I have some questions that should be answered, after which I believe that it can be published.

1. Does the formation of the oxide surface layer subsequently inhibit Li insertion in later cycles? Li diffusion inside iron fluorides often has a lower activation barrier along rapid transport directions than the barrier for Li to enter the crystal surface. Is there a high surface barrier in the FeCoOF crystal (using the lowest surface energy surface, as this would be most prevalent)? Is this surface barrier affected by the oxide that is believed to form at the surface?
2. Related to question 1 regarding the formation of an oxide surface film, F loss under the beam in

TEM has been known to occur. Are the authors sure that the implication of O-enriched surface is not an artifact of the loss of F at the surface?

3. In consideration of their comments on pg 15, line 317, is their statement compromised by their figure S24e: In that figure, at 5-20 nm the F is highly depleted in comparison to the O.

4. Pg 18, line 366: Intercalation has been seen to occur before conversion in the Fe-fluorides, but have been rebuffed by other studies. Does intercalation actually occur before extrusion, or is extrusion a surface event that moves inward with the Li?

Detailed responses to the reviewers' comments

Reviewer #1 (Remarks to the Author):

Manuscript NCOMMS-17-30638

The paper by Xiulin Fan et al entitled “High Energy-Density and Reversibility of Iron Fluoride Cathode Enabled Via an Intercalation-Extrusion Reaction” is a well written paper dealing with synthesis, characterization and electrochemical performance of Fe_{0.9}Co_{0.1}OF based cathodes. This paper follows in the steps of past developments on iron fluorides cathodes from FeF₂ to FeF₃ and FeOF which show continuing improvements in cycle life. In this paper, partial substitution of Fe by Co gives rise to further improvements in cycling performance and voltage hysteresis of iron fluoride cathodes. The results reported in this paper are excellent with detailed TEM and PDF characterization techniques and should be of interest to scientists in the energy storage field. This paper can be published in the present form with minor corrections. A few general comments:

***Reply:** We really appreciate the positive comment on our paper.*

1. Mixed metal fluorides of composition Cu_yFe_{1-y}F₂ were previously investigated by Wang et al (doi:10.1038/ncomms7668) (Ref 14 in the present paper) where they show that with Cu substitution, they can achieved very low hysteresis (over potential) of less than 150 meV. Reference to this paper should be mentioned in the section on over potential starting with line 192 as this work is complementary to the Co substitution presented in this study which reports a decrease in voltage hysteresis of Fe_{0.9}Co_{0.1}OF as compared to FeOF.

***Reply:** Thanks for the constructive suggestions.*

According to the comment, we have added this in the revised manuscript:

The over-potential in $\text{Fe}_{0.9}\text{Co}_{0.1}\text{OF}$ (170 mV) is also significantly smaller than the reaction in FeOF (286 mV) and FeF_3 (634 mV) (Figure S13), and comparable to the lowest over-potential ever reported in mixed metal fluoride cathode ($\text{Cu}_y\text{Fe}_{1-y}\text{F}_2$, 150 mV).¹⁷

2. Too many sub-figures are placed within single figures making them too small and totally unreadable. My recommendation is to increase the number of figures and enlarge their sizes in a way to make the lettering size as recommended by the publisher. This is especially true for Figure 2.

***Reply:** We appreciate the constructive comment.*

According to the comment, we increased the number of the figures and enlarged the size of every figure in Figure 2. And we put the inset-figure in Figure 2a into the supporting information as Figure S9. Besides, we also enlarged the figures in Figure 3.

3. In Figure caption 4, please specify if the elemental distribution is obtained by STEM-EELS or EDX techniques.

***Reply:** We obtained the elemental distribution using STEM-EELS. We add “(e) obtained by STEM-EELS technique” into the caption for Figure 4*

Reviewer #2 (Remarks to the Author):

This is a very interesting paper in which the authors have done a significant number of experiments to evaluate the effect of adding Co to FeOF in order to change the

deleterious hysteresis between charge and discharge. By removing the conversion reactions, which form Fe crystals in the conversion fluorides, the hysteresis appears to be significantly reduced. The work provides a new route to a higher capacity cathode and improves the previously considered Fe-fluorides and oxy-fluorides. Their compositional modifications may open the field to other additives that may also show further improvements. The paper needs some editing in wording, and I have some questions that should be answered, after which I believe that it can be published.

Reply: We really appreciate the positive comment from reviewer.

1. Does the formation of the oxide surface layer subsequently inhibit Li insertion in later cycles? Li diffusion inside iron fluorides often has a lower activation barrier along rapid transport directions than the barrier for Li to enter the crystal surface. Is there a high surface barrier in the FeCoOF crystal (using the lowest surface energy surface, as this would be most prevalent)? Is this surface barrier affected by the oxide that is believed to form at the surface?

Reply: We thank the insightful comments from reviewer. As shown in our manuscript, the formation of an oxide surface layer with the disordered rocksalt structure with O vacancies can improve the stability of the structure of the cathode during cycling.

This thin layer with thickness of ~ 2 nm will not inhibit the Li insertion in later cycles according to our DFT energy barrier calculation. The main component for the oxide surface layer is considered to be rocksalt LiFeO₂ according to our results shown in the manuscript. To understand the Li diffusion in the LiFeO₂, the energy barrier is

calculated using climbing image nudged elastic band (CINEB) method implemented in VASP.¹ The computation details were added in the supporting information. According to the symmetry of LiFeO_2 , only one diffusion path is considered. The calculated diffusion energy barriers and its corresponding diffusion path are plotted in Figure R1b. According to the results, Li diffuses from the octahedral to the tetrahedral site where the transition state lies on. The diffusion barrier is calculated to be 0.29 eV. This diffusion barrier is similar to that in LiFePO_4 which is considered to be a high rate cathode.²

Figure R1. (a) Illustration of rocksalt LiFeO_2 , the orange and green octahedral represent the FeO_6 and LiO_6 structure, respectively; (b) Calculated Li diffusion energy barriers and its corresponding diffusion path in LiFeO_2 using CINEB method.

A high surface energy barrier is found in many other cathode materials. However, the high surface energy barrier has little impact on Li diffusion in our FeCoOF system according to the DFT calculation results. To investigate the effect of surface energy, the slab structure of FeCoOF with a max Miller index of 1 is generated by Pymatgen.^{3,4} According to the Task's classification,⁵ only the stoichiometric surfaces with a vanishing

dipole in the direction of the surface normal is considered in our calculation as listed in **Error! Reference source not found.** To calculate the surface energy, we use slabs more than 15 Å and a 12 Å vacuum is used for each model. The outmost five atom layers of each surface of the slabs are relaxed while the middle layers are fixed to model as bulk. The most energy favorable surface is (100) surface with F as the terminals, as shown in Figure R2.

Table R1. The calculated surfaces with different Miller index and terminals and their corresponding surface energies.

Surface	100		110		
	1	2	1	2	3
Surface energy (J/m ²)	0.55	0.28	0.94	0.55	0.82

To assess the surface to bulk migration barrier, the (001) surface terminated by F layer and having a vacuum of 12 Å is used as it is calculated to have the lowest energy. The relative energy of Li ion migration from the surface to the bulk-like slab center via O-channel and F-channel are calculated, as shown in Figure R2a, b. Constrained minimization is used to hold the depth of the Li ion fixed while all other degrees of freedom are relaxed, as the previous publication.⁶ The results are shown in Figure R2c. It can be found that F-channel is more suitable for the Li migration. The diffusion barrier at the surface is about 0.88 eV. However, the diffusion energy barrier for the bulk is calculated to be 1.36 eV which is higher than the surface diffusion barrier. Therefore, the surface energy barrier effect is not a key problem for the FeCoOF system.

Figure R1 Considered diffusion path from the surface to the bulk via (a) the O-channel (b) F-channel and (c) their calculated relative energies.

In the revised manuscript, we added:

It should be noted that this O-rich oxide layer with thickness of ~ 2 nm will not retard the Li insertion in later cycles according to our DFT energy barrier calculations (Figure S27). The diffusion barrier in LiFeO_2 rocksalt phase is calculated to be 0.29 eV, which is similar to that in LiFePO_4 cathode materials.⁵⁰

First, it should be pointed out that similar to the FeF_3 , the surface energy barrier has minor impact on the Li diffusion in the $\text{Fe}_{0.9}\text{Co}_{0.1}\text{OF}$ system according to the DFT calculation results (Figure S28 and Table S1). The surface diffusion barrier (0.88 eV) is lower than the bulk diffusion barrier (1.36 eV) in $\text{Fe}_{0.9}\text{Co}_{0.1}\text{OF}$.

We supplemented the Figure R1 and R2, and Table R1 in the revised Supporting Information (Figure S27, Figure S28, and Table S1), and also supplemented the related discussions in the revised Supporting Information.

2. Related to question 1 regarding the formation of an oxide surface film, F loss under the beam in TEM has been known to occur. Are the authors sure that the implication of O-enriched surface is not an artifact of the loss of F at the surface?

***Reply:** We agree with the reviewer that electron beam will cause the loss of F, especially in the case of STEM-EELS 2D mapping which takes a longer time. We are aware of beam damage issue when we did STEM-EELS 2D mapping. We measured surface F contents of both cycled FeOF and $Fe_{0.9}Co_{0.1}OF$ using STEM-EELS line scan at different dwelling times. The shorter time we used the smaller damage the electron beam causes. Although the nano-phase separations were detected in the cycled FeOF, our results show that surface F content in FeOF is high while it is very low in $Fe_{0.9}Co_{0.1}OF$ even at the beginning of STEM-EELS 2D mapping. Moreover, we did not detect the loss of F at the surface for the pristine $Fe_{0.9}Co_{0.1}OF$ materials. Therefore, oxygen-rich surface in the cycled $Fe_{0.9}Co_{0.1}OF$ is intrinsic and not an artifact caused by electron beam damage.*

3. In consideration of their comments on pg 15, line 317, is their statement compromised by their figure S24e: In that figure, at 5-20 nm the F is highly depleted in comparison to the O.

***Reply:** We appreciate this comment. We have revised the statement on FeOF on page 15, line 317.*

In the revised manuscript, we replaced the statements on page 15, line 317 by following statement:

However, no such layer was observed in FeOF. Instead, nano-sized domains with size of 10-20 nm were observed, where significant element segregations can be detected as shown in Figure S25.

For $Fe_{0.9}Co_{0.1}OF$, it can be seen that a surface layer almost F-free is formed after prolonged cycling. The thickness of this layer is about 2 nm (Figure 4, and Figure S23c). Apart from this thin layer, the O and F elements are homogeneously distributed in the materials (as shown in Figure 4 and S23c)

However, for the FeOF material, no such layer can be observed. Instead, obvious nano-phase separation is observed, as denoted by the dash red lines in Figure S24b and c, and also can be seen in Figure S24g, 24h, and 24i. This nano-phase (with size of 10-20 nm) separation results in the separation of the O and F, which is obviously detected in the elemental mapping results (Figure S24g, 24h, and 24i). This nano-phase separation results in the O-rich area and F-rich area. It can be seen that for the Figure 24e “surface area” with thickness of about 15 nm are O-rich phase (the position is between 0-22 nm), while the center area is F-rich area (the position is from 35 nm to the center in Figure S24e). Therefore, this F depletion in some area is due to the nano-phase separation not due to the surface layer.

4. Pg 18, line 366: Intercalation has been seen to occur before conversion in the Fe-fluorides, but have been rebuffed by other studies. Does intercalation actually occur before extrusion, or is extrusion a surface event that moves inward with the Li?

Reply: *We appreciate for the comment.*

The calculation of the lithiation process is based on the thermodynamic data, which shows that all levels of lithiation should be “conversion phases” because of the lowest energies as shown in the blue line in Figure 5c and Figure S27. However, given that the small energy difference between the intercalation (red line in Figure 5c) and the conversion reaction (blue line in Figure 5c) and large kinetic barriers to the conversion reactions, it is reasonable that the material intercalate Li before it converts, which is indeed inconsistent with the experiment.⁷ It should be pointed out that Ceder et al also found this in-consistency between the calculation and the experimental in FeOF system.⁷

For FeF_3 and FeOF (or $\text{FeO}_x\text{F}_{1-x}$) materials, before conversion reaction, intercalation reaction will occur, which has been proved by PDF,⁸ XRD,^{9,10} and NMR.⁹ The materials based on the intercalation reaction show a good cycling performance and rate performance. However, once the conversion reaction was initiated, the cycling performance, rate capability, and hysteresis deteriorated immediately and seriously.¹¹ Therefore, to get a good cycling performance for these materials, previous studies mainly focused on the intercalation reactions by discharging the cells with a higher cutting-off voltage.^{11,12}

Therefore, based on the above analysis, we can conclude that the intercalation will occur before the conversion reaction in the Fe-fluorides, and it is not an extrusion reaction that moves inward with the Li.

It is worth mentioning that such reaction mechanism is not always adopted for all of the fluoride systems. For example, Wang et al (Nature Commun., 6, 6668, doi:10.1038/ncomms7668) showed that there was no intercalation reaction in the $\text{Cu}_y\text{Fe}_{1-y}\text{F}_2$ materials (complex agglomerates of small nanocrystallites made by

mechanochemical synthesis method). These results suggest that the nature of reaction mechanism can be dependent on transition metal chemistries. Such dependence in fact provides possibility of tuning the reaction route through elemental doping and tailoring the synthesis route, which is exactly the motivation of this work.

In the revised manuscript, we added:

It is worth mentioning that such reaction mechanism does not always apply for all fluoride systems. For example, Wang et al¹⁷ showed that there was no intercalation reaction in the $\text{Cu}_y\text{Fe}_{1-y}\text{F}_2$ materials, suggesting that the nature of reaction mechanism depends on transition metal chemistries. Such dependence in fact provides possibility of tuning the reaction route through elemental doping and tailoring the synthesis route, which is exactly the motivation of this work.

References

- 1 Henkelman, G., Uberuaga, B. P. & Jónsson, H. A climbing image nudged elastic band method for finding saddle points and minimum energy paths. *The Journal of chemical physics* **113**, 9901-9904 (2000).
- 2 Dathar, G. K. P., Sheppard, D., Stevenson, K. J. & Henkelman, G. Calculations of Li-Ion Diffusion in Olivine Phosphates. *Chemistry of Materials* **23**, 4032-4037, doi:10.1021/cm201604g (2011).
- 3 Sun, W. & Ceder, G. Efficient creation and convergence of surface slabs. *Surface Science* **617**, 53-59, doi:10.1016/j.susc.2013.05.016 (2013).
- 4 Kamrani Moghaddam, L., Ramezani Pascheperi, S., Zaimy, M. A., Abdalaian, A. & Jebali, A. The inhibition of epidermal growth factor receptor signaling by hexagonal selenium nanoparticles modified by SiRNA. *Cancer gene therapy* **23**, 321-325, doi:10.1038/cgt.2016.38 (2016).
- 5 Tasker, P. The stability of ionic crystal surfaces. *Journal of Physics C: Solid State Physics* **12**, 4977 (1979).
- 6 Tompsett, D. A., Parker, S. C., Bruce, P. G. & Islam, M. S. Nanostructuring of $\beta\text{-MnO}_2$: The Important Role of Surface to Bulk Ion Migration. *Chemistry of Materials* **25**, 536-541, doi:10.1021/cm303295f (2013).
- 7 Chevrier, V. L., Hautier, G., Ong, S. P., Doe, R. E. & Ceder, G. First-principles study of iron oxyfluorides and lithiation of FeOF. *Physical Review B* **87**, 094118 (2013).
- 8 Wiaderek, K. M. *et al.* Comprehensive Insights into the Structural and Chemical Changes in Mixed-Anion FeOF Electrodes by Using Operando PDF and NMR Spectroscopy. *Journal of the American Chemical Society* **135**, 4070-4078, doi:10.1021/ja400229v (2013).

- 9 Yamakawa, N., Jiang, M., Key, B. & Grey, C. P. Identifying the Local Structures Formed during Lithiation of the Conversion Material, Iron Fluoride, in a Li Ion Battery: A Solid-State NMR, X-ray Diffraction, and Pair Distribution Function Analysis Study. *Journal of the American Chemical Society* **131**, 10525-10536, doi:10.1021/ja902639w (2009).
- 10 Badway, F., Cosandey, F., Pereira, N. & Amatucci, G. G. Carbon Metal Fluoride Nanocomposites: High-Capacity Reversible Metal Fluoride Conversion Materials as Rechargeable Positive Electrodes for Li Batteries. *J Electrochem Soc* **150**, A1318-A1327, doi:10.1149/1.1602454 (2003).
- 11 Tan, H. J. *et al.* Electrochemical Cycling and Lithium Insertion in Nanostructured FeF₃ Cathodes. *J Electrochem Soc* **161**, A445-A449, doi:10.1149/2.096403jes (2014).
- 12 Ma, D.-l. *et al.* Three-dimensionally ordered macroporous FeF₃ and its in situ homogenous polymerization coating for high energy and power density lithium ion batteries. *Energy & Environmental Science* **5**, 8538-8542, doi:10.1039/C2EE22568A (2012).

REVIEWERS' COMMENTS:

Reviewer #2 (Remarks to the Author):

the revised paper is acceptable now.

Reviewers' comments:

Reviewer #2 (Remarks to the Author):

The revised paper is acceptable now.

Reply: *We really appreciate the positive comment on our paper. Thanks.*